# Diffusive excitonic bands from frustrated triangular sublattice in a singlet-ground-state system

Bin Gao [1,12], Tong Chen [1,11,12], Xiao-Chuan Wu [2], Michael Flynn [3,4], Chunruo Duan[1], Lebing Chen [1], Chien-Lung Huang[1,5], Jesse Liebman [1,11], Shuyi Li[1], Feng Ye [6], Matthew B. Stone [6], Andrey Podlesnyak [6], Douglas L. Abernathy [6], Devashibhai T. Adroja [7], Manh Duc Le [7], Qingzhen Huang[8], Andriy H. Nevidomskyy [1], Emilia Morosan[1], Leon Balents[9,10] & Pengcheng Dai [1] ✉

Magnetic order in most materials occurs when magnetic ions with finite moments arrange in a particular pattern below the ordering temperature. Intriguingly, if the crystal electric field (CEF) effect results in a spin-singlet ground state, a magnetic order can still occur due to the exchange interactions between neighboring ions admixing the excited CEF levels. The magnetic excitations in such a state are spin excitons generally dispersionless in reciprocal space. Here we use neutron scattering to study stoichiometric $Ni_2Mo_3O_8$, where $Ni^{2+}$ ions form a bipartite honeycomb lattice comprised of two triangular lattices, with ions subject to the tetrahedral and octahedral crystalline environment, respectively. We find that in both types of ions, the CEF excitations have nonmagnetic singlet ground states, yet the material has magnetic order. Furthermore, CEF spin excitons from the tetrahedral sites form a dispersive diffusive pattern around the Brillouin zone boundary, likely due to spin entanglement and geometric frustrations.

In most magnetic materials, the ground state of electron orbital states of magnetic ions in the crystal electric field (CEF) produced by the surrounding charge anion neighbors is magnetic with a finite moment and nonzero spin[1,2]. The long-range magnetic ordered structure and ordering temperature are determined by the magnetic exchange interactions ($J$), and the ordered moment direction is controlled by (typically small) easy-axis anisotropy gap[2]. However, there are also materials where the ground state of CEF levels of magnetic ions is a nonmagnetic singlet. Here, magnetic ordering and properties are sensitive to the ratio of magnetic exchange $J$ to single-ion anisotropy (SIA, $D$) which is controlled by the CEF energy level of the first excited state[3–5]. With negligible magnetic exchange ($J \ll D$), the system is paramagnetic at all temperatures. For a relatively large magnetic exchange ($J \gg D$), magnetic order will be induced through a polarization instability of the singlet-ground state, termed an induced moment[3–8]. In this case, the basic magnetic excitations that describe

[1]Department of Physics and Astronomy, Rice University, Houston, TX 77005, USA. [2]Department of Physics, University of California, Santa Barbara, CA 93106, USA. [3]Department of Physics, University of California, Davis, CA 95616, USA. [4]Department of Physics, Boston University, Boston, MA 02215, USA. [5]Department of Physics and Center for Quantum Frontiers of Research & Technology (QFort), National Cheng Kung University, 701 Tainan, Taiwan. [6]Neutron Scattering Division, Oak Ridge National Laboratory, Oak Ridge, TN 37831, USA. [7]ISIS Neutron and Muon Source, Rutherford Appleton Laboratory, Chilton, Didcot OX11 0QX, UK. [8]NIST Center for Neutron Research, National Institute of Standards and Technology, Gaithersburg, MD 20899, USA. [9]Kavli Institute for Theoretical Physics, University of California, Santa Barbara, CA 93106, USA. [10]Canadian Institute for Advanced Research, Toronto, ON, Canada. [11]Present address: Department of Physics and Astronomy, Johns Hopkins University, Baltimore, MD 21218, USA. [12]These authors contributed equally: Bin Gao, Tong Chen. ✉e-mail: pdai@rice.edu

the CEF transitions propagating through the lattice are called spin excitons[3–7], analogous to electronic excitons that are bound states of an electron and a hole in a solid[9]. Spin excitons are fundamentally different from spin waves (magnons), which are strongly dispersive collective modes associated with spin precession on the lattice of magnetically ordered materials, and which disappear above the magnetic ordering temperature for isotropic Heisenberg magnets. In most cases, spin excitons originate from CEF levels of a localized single ion. Therefore, they are expected to be dispersionless in reciprocal space and well defined in both the magnetically ordered and paramagnetic states. However, when dispersive spin excitons are observed, the dispersion of these excitations can reveal unique information concerning magnetic exchange interactions between the localized ionic sites (spin–spin entanglement) and their relationship with the magnetically ordered state[3–7,10,11].

Here we use thermodynamic and neutron scattering experiments to study stoichiometric honeycomb lattice antiferromagnetic (AF) ordered magnet $Ni_2Mo_3O_8$, where $Ni^{2+}$ ions form a bipartite honeycomb lattice comprised of two triangular lattices, in the tetrahedral and octahedral crystalline environment, respectively (Fig. 1a–c)[12,13]. We find that CEF levels of $Ni^{2+}$ ions from both tetrahedral and octahedral environments have a nonmagnetic spin-singlet-ground state but with very different single-ion anisotropy energy scales for the two sites. Spin excitations of CEF levels (spin excitons) from the $Ni^{2+}$ triangular tetrahedral sites form a diffusive pattern around the Brillouin zone (BZ) boundary in the AF and paramagnetic states in momentum space. Therefore, $Ni_2Mo_3O_8$ realizes a novel situation in which the exchange energy falls between two very different single-ion energies, leading to a cooperative mechanism for magnetic order and strongly dispersing excitons associated with the larger single-ion anisotropy of the tetrahedral sites. Due to this hierarchy of energy scales, the excitons can persist even when the magnetic order is destroyed above the Néel

temperature. In this regime, the excitons are strongly scattered from spin fluctuations and give rise to a distinct mechanism for dispersive diffusive scattering, hosting a unique coexistence of heavy particles (tetrahedral excitons) propagating in a frustrated background of light but dense (octahedral) spins due to spin entanglement and geometric frustrations.

## Results

### Crystal structure, magnetic order, susceptibility, and specific heat of $Ni_2Mo_3O_8$

The $M_2Mo_3O_8$ (M = Fe, Mn, Ni, Co, Zn) compounds have drawn increasing attention due to their multiferroic properties[12–16]. The crystal structure of $M_2Mo_3O_8$ consists of magnetic bipartite honeycomb M-O layers, separated by sheets of $Mo^{4+}$ layers (Fig. 1a), where the $Mo^{4+}$ ions inside each layer are trimerized and form a singlet. The two $M^{2+}$ sites have different oxygen coordination, with one site being an $MO_6$ octahedron and the adjacent one being an $MO_4$ tetrahedron. In this family, $Ni_2Mo_3O_8$ was studied as a platform to explore the physics of geometrically frustrated lattice[12,13]. Neutron powder diffraction experiments reveal that both the $MO_6$ octahedron and the $MO_4$ tetrahedron each form perfect triangular lattices with no inter-site disorder, and the system orders antiferromagnetically with a Néel temperature of $T_N = 5.5\,K$[12,13]. The magnetic structure is stripe like within the Ni-O plane, and zig-zag like along the c-axis with different ordered moments for octahedral and tetrahedral Ni sites (Fig. 1c and d). Previous single-ion CEF analysis suggests that both octahedral and tetrahedral Ni sites have nonmagnetic singlet-ground states, and the first excited magnetic doublets are at 7.8 and 23 meV, respectively[12].

Even though the Ni ions appear to form a simple (bipartite) honeycomb lattice in $Ni_2Mo_3O_8$ (Fig. 1b–d), this compound should be viewed rather as two inter-penetrating triangular lattices formed by $NiO_6$ octahedron and $NiO_4$ tetrahedron. In $Ni_2Mo_3O_8$, magnetic order

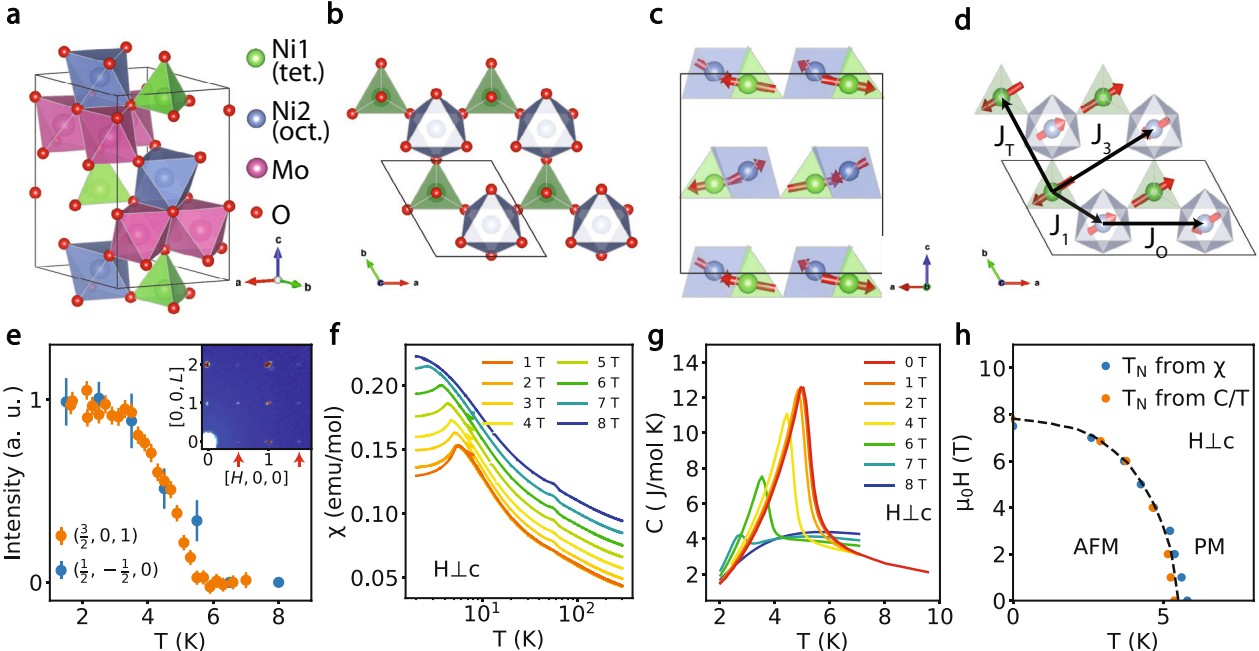

**Fig. 1 | Structure, Brillouin zone, magnetic Bragg peak, d.c. susceptibility and specific heat of $Ni_2Mo_3O_8$. a** A schematic of the structure of $Ni_2Mo_3O_8$ with a-, b-, c-axis labeled. **b** The octahedral coordinated Ni ions (blue) and the tetrahedral coordinated Ni ions (green) form a quasi-2D honeycomb lattice. **c, d** Side (along the b-axis) and top (along the c-axis) view of the magnetic structure of $Ni_2Mo_3O_8$. The black frames indicate the size of magnetic unit cell and black arrows denote magnetic exchange couplings along different directions. **e** Temperature dependence of the intensities in arbitrary units (a. u.) of two magnetic Bragg peaks at (3/2,0,1) and (1/2,−1/2,0). Inset shows a 2D color plot in the [H, 0, L] plane, in which

magnetic Bragg peaks are indicated by red arrows. The vertical error bars are statistical errors of 1 standard deviation. **f** The d.c. susceptibility with 1–8 T magnetic fields perpendicular to the c-axis. Data are shifted by magnetic field × 0.01 emu / mol T for clarity. Magnetic transition is suppressed in the 8 T field. **g** The magnetic contribution to the specific heat $C_{mag}$ with 0-8 T magnetic fields perpendicular to the c-axis. Magnetic transition is suppressed in the 8 T field.
**h** Temperature versus magnetic field phase diagram from d.c. susceptibility and specific data.

**Table 1 | Positions of atoms in $Ni_2Mo_3O_8$ as determined from single crystal X-ray diffraction**

|       | x         | y         | z          | U           | Occ        |
|-------|-----------|-----------|------------|-------------|------------|
| Ni(1) | 1/3       | 2/3       | 0.94918(9) | 0.00527(16) | 0.9797(97) |
| Ni(2) | 1/3       | 2/3       | 0.51172(9) | 0.00426(15) | 0.993(99)  |
| Mo    | 0.14617(2)| 0.85383(2)| 0.24966(3) | 0.00342(9)  | 0.985(98)  |
| O(1)  | 0         | 0         | 0.3920(4)  | 0.0051(7)   | 1          |
| O(2)  | 0.3333    | 0.6667    | 0.1466(5)  | 0.0051(7)   | 1          |
| O(3)  | 0.4883(3) | 0.5117(3) | 0.3671(3)  | 0.0053(4)   | 1          |
| O(4)  | 0.1692(3) | 0.8308(3) | 0.6334(4)  | 0.0055(4)   | 1          |

Single crystal X-ray diffraction refinement on $Ni_2Mo_3O_8$. Over 15,000 Bragg peaks are collected and refined with a space group $P6_3mc$. The positions and occupation fractions are refined, yielding no magnetic/nonmagnetic disorder. The fitting results an R1 = 1.03%.

is assumed to arise from the relatively large magnetic exchange coupling between the octahedral and tetrahedral Ni sites (denoted as $J_1$ in Fig. 1d) in comparison with the energy of the CEF level of the Ni octahedral site[12]. However, there are no inelastic neutron scattering (INS) experiments to date to identify the CEF levels of the Ni octahedral and tetrahedral sites, and prove that the ground state is indeed a singlet. Here we report magnetic susceptibility, heat capacity, X-ray diffraction, and INS experiments on single crystals of $Ni_2Mo_3O_8$ and $NiZnMo_3O_8$ grown by the chemical vapor transfer method. Our careful single crystal X-ray diffraction experiments reveal that both the $NiO_6$ octahedra and $NiO_4$ tetrahedra form perfect triangular lattices with no inter-site disorder, and there is also no disorder on the Mo site (see Table 1 and Supplementary Information for details). The X-ray and neutron diffraction refinements also show that nonmagnetic dopant ions like $Zn^{2+}$ prefer to occupy the tetrahedral sites[12]. Consistent with previous neutron powder diffraction work[12,13], our neutron single crystal diffraction refinements find that the spins of $Ni^{2+}$ ions form a stripy AF order (Fig. 1c and d) below $T_N = 5.5$ K (Fig. 1e), but with the ordered moments of tetrahedral and octahedral Ni sites being 1.47 and 1.1 $\mu_B$, respectively, different from the previously reported values.

Figure 1f shows the temperature dependence of the d.c. susceptibility $\chi(T)$ with 1–8 T magnetic fields applied along the [1,1,0] direction perpendicular to the $c$-axis. The data in the 1 T field (orange line) shows a clear peak around $T_N \approx 6$ K consistent with neutron data in Fig. 1e, while the data in the 8 T field (blue line) shows no evidence of a magnetic transition. In-plane susceptibility is much larger than the $c$-axis susceptibility (see Supplementary Fig. 3), indicating easy-plane anisotropy. The Curie-Weiss fitting to the in-plane and $c$-axis susceptibility above 200 K gives $\theta_{CW\perp} = -454.12 \pm 0.53$ K and $\theta_{CW,//} = -100.52 \pm 0.11$ K, respectively, suggesting a quasi-2D system where the in-plane magnetic exchange is larger than the $c$-axis exchange. The difference between our analysis and previous results on single crystals mainly comes from the direction of the in-plane magnetic fields[13]. Figure 1g shows the temperature dependence of the magnetic contribution to the specific heat as a function of applied magnetic fields perpendicular to the $c$-axis. At zero field (red line), we see a typical λ-shaped transition around $T_N \approx 6$ K. At 8 T field, there is no evidence of a magnetic transition above 2 K, consistent with Fig. 1f. Figure 1h shows the temperature-field phase diagram from our susceptibility and specific heat data.

## Neutron scattering studies of spin waves and CEF levels

To search for the CEF levels of the tetrahedral and octahedral Ni sites and demonstrate that the ground state of $Ni_2Mo_3O_8$ is indeed a spin singlet, we carried out INS experiments on single crystalline and powdered samples with incident energies ($E_i$) of 2.5 meV, 3.7 meV, 40 meV, 250 meV, and 1.5 eV. At $E_i = 2.5$ meV, we see clear dispersive spin waves at 1.7 K (Fig. 2a) with two modes. This is consistent with the expectations from the linear spin-wave theory (LSWT) calculation,

since there are four $Ni^{2+}$ sites in the magnetic unit cell, resulting in two doubly-degenerate modes. The small anisotropic spin gap (<0.3 meV) and overall spin-wave energy bandwidth of 1.5 meV are consistent with thermal dynamic data in Fig. 1f–h and $T_N = 5.5$ K. Figure 2b and c shows a powder averaged INS $S(E,\mathbf{Q})$ spectrum, where $E$ and $\mathbf{Q}$ are energy and moment transfer, respectively, and a constant-$|\mathbf{Q}|$ cut with $E_i = 40$ meV at temperatures below and above $T_N$. At 1.7 K, there are at least three excitation peaks from the $Ni^{2+}$ spins at 13.8, 16.9, and 20.3 meV, and no visible excitations from 2 to 10 meV and above 40 meV. These excitations cannot be spin waves since there are already two modes below 1.5 meV. They must arise from the single-ion CEF levels which we analyze using the Stevens operator formalism, in which, due to the $C_{3V}(3m)$ point-group symmetry, both tetrahedral and octahedral Ni sites are described by the Hamiltonian $H_{CEF} = B_2^0 \hat{O}_2^0 + B_4^0 \hat{O}_4^0 + B_4^3 \hat{O}_4^3$, where $B_2^0$ and $B_4^m$ are the second- and fourth-order crystal field parameters and $\hat{O}_2^0$ and $\hat{O}_4^m$ are the corresponding Stevens operators. As described in ref. 12, the orbital ground state is the $^3A$ state which has a three-fold degeneracy that is further lifted by the spin-orbit coupling (SOC) producing a singlet and doublet (Fig. 2d). Since the $Ni^{2+}$ ions in $NiZnMo_3O_8$ tend to occupy octahedral sites, we performed INS experiments on the powder sample of $NiZnMo_3O_8$ to study the single-ion crystal fields of octahedral sites[12]. We find that the intensity of the scattering in the range of 12–21 meV is considerably reduced (see Supplementary Fig. 4), consistent with the percentage of the Ni in the tetrahedral site of $NiZnMo_3O_8$ determined from neutron powder diffraction. For comparison, the spin excitations in $NiZnMo_3O_8$ are mostly centered below 2 meV, consistent with the notion that the energy of the CEF doublet levels from the octahedral site is below 2 meV (see Supplementary Fig. 5), which is clearly different from earlier low resolution electron spin resonance measurements and estimation from the point charge model[12]. Therefore, the three CEF levels observed in $Ni_2Mo_3O_8$ at 1.7 K near 17 meV are all from the first excited doublet of the tetrahedral Ni site. Since the magnetic unit cell doubles the structural unit cell in the ordered state and the molecular field from the ordered moments splits the doublets, one would expect up to four excitation modes from the Ni tetrahedral site below $T_N$, while there is only one excitation above $T_N$. On cooling below $T_N$ from 10 K, we see a clear splitting of the broad CEF peak at ~17 meV into two peaks around 13 meV and a broad peak around 20 meV, consistent with this picture (Fig. 2c and d). Combined with susceptibility data in Fig. 1f, we construct the Ni CEF levels as shown in Fig. 2d. While both Ni sites have singlet-ground states, the first excited state for the Ni tetrahedral and octahedral sites is at ~17 and ~1 meV, respectively. Since the energy bandwidth of spin waves in the AF ordered $Ni_2Mo_3O_8$ is less than 2 meV (Fig. 2a), we estimate that the magnetic exchange interactions between the Ni tetrahedral and octahedral sites $J_1$, and second-neighbor tetrahedral (octahedral) and tetrahedral (octahedral) sites $J_T$ ($J_O$) to be less than 2 meV (Fig. 1d). Therefore, we identify $Ni_2Mo_3O_8$ as a spin-singlet-ground-state system with magnetic order being induced by the exchange $J_1$ comparable to the CEF level of the Ni octahedral site ($J_1 > D_O \approx 0.8$ meV, Fig. 2e and f). For comparison, we note that $Co_2Mo_3O_8$ has a much higher $T_N$ ~40 K[17] and its upper band of spin waves is around 12 meV[18].

To determine the energy, wavevector, and temperature dependence of the Ni tetrahedral excitonic magnetism in $Ni_2Mo_3O_8$, we co-aligned high-quality single crystals in the $[H, H, 0] \times [-K, K, 0]$ scattering plane (Fig. 3a and b). Figure 3c–e is the $E$-$\mathbf{Q}$ dispersions of the spin excitons at 1.5, 10, and 120 K. We observed magnetic scattering in two separated energy regions, 12–16 and 18–22 meV, at 1.5 K. Above $T_N$, the scattering below and above 17 meV merge and become dispersive. The dispersion persists up to 120 K, which is one of the signatures of excitons. Figure 3f–k shows reciprocal space maps of the spin excitations in the $[H, K, 0]$ plane in the two energy regions at 1.5, 10, and 120 K. The maps at different temperatures show qualitatively the same features: For $E = 20 \pm 2$ meV, the scattering show broad peaks

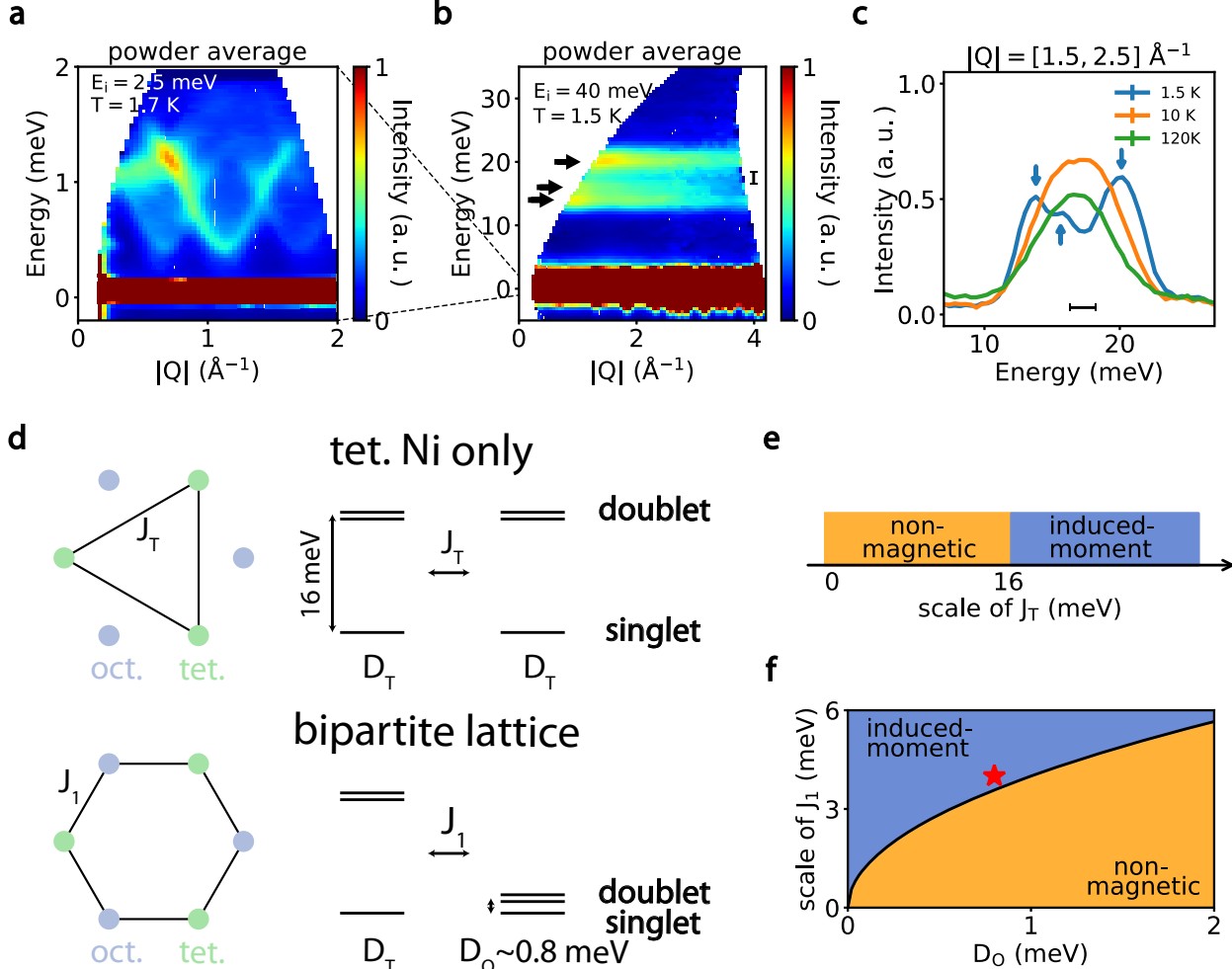

**Fig. 2 | Powder inelastic neutron spectra and crystal field levels of Ni$_2$Mo$_3$O$_8$.** **a** Powder average of the low-energy spin waves of Ni$_2$Mo$_3$O$_8$ single crystal at 1.7 K with $E_i$ = 2.5 meV, which cannot be resolved in the spectrum using $E_i$ = 40 meV. **b** Powder average of the spin excitation spectrum of Ni$_2$Mo$_3$O$_8$ single crystal at 1.5 K with $E_i$ = 40 meV. Black arrows denote the three excitonic levels at 13.8, 16.9, and 20.3 meV. **c** Temperature dependence of the constant-$Q$ ([1.5–2.5] Å$^{-1}$) cuts at 1.5 K from data in panel **b**. Blue arrows denote three peaks at 13.8, 16.9, and 20.3 meV. The horizontal bar is the energy resolution. **d** Schematics of crystal field levels of the tetrahedral coordinated and the octahedral coordinated Ni ions. **e, f** Comparison of the magnetic exchange couplings and CEF energy levels for Ni ions in tetrahedral and octahedral sites, respectively. One expects a nonmagnetic ground state for tetrahedral site while a magnetic ordered state is expected in the octahedral sites.

centered near the Brillouin zone center Γ points (Fig. 3f–h), while for $E$ = 14 ± 2 meV, the scattering is like the complementary part of the high energy scattering that forms a diffusive pattern around the zone boundary (Fig. 3i–k). In both cases, the scattering below and above $T_N$ are similar, contrary to the expected broadening of spin waves in momentum space from a magnetically ordered state to a paramagnetic state across $T_N$. Since spin waves in Ni$_2$Mo$_3$O$_8$ have a band top of ~1.5 meV at 1.7 K (Fig. 2a), the broad dispersive excitations in Fig. 3 f and i at 1.5 K well below $T_N$ cannot arise from spin waves of magnetic ordered Ni$^{2+}$.

Figure 3l and m shows the in-plane magnetic field dependence of $S(E,Q)$ at energies near the CEF levels of Ni tetrahedron at 2 K for zero and 5-T field-polarized ferromagnetic state, respectively. The wavevector dependence of spin excitations for zero (Fig. 3n) and 5-T (Fig. 3o) field at $E$ = 20 ± 2 meV and 2 K are similar to zero field data at 1.5 K (Fig. 3f) and 10 K (Fig. 3g), respectively. The situation is similar at $E$ = 14 ± 2 meV (Fig. 3i at 1.5 K and 3j at 10 K in zero field, Fig. 3p at zero field and 3q at 5-T both at 2 K). This reflects the fact that the paramagnetic and ferromagnetic states have the same periodicities and that the exchange interaction does not appreciably change with field or temperature, whereas the AF state has a different

periodicity, as the magnetic unit cell is doubled that of the structural cell in the AF phase. However, the paramagnetic and ferromagnetic states differ in that the applied field splits the excited doublet in the ferromagnetic state while there is no splitting in the paramagnetic state. This splitting is not seen in the data because it is smaller (≈0.1 meV) than the instrument resolution (≈1 meV), whereas the splitting due to the magnetic ordering is expected to be an order of magnitude larger than that produced by the external field.

## Discussion

There are several unusual features in the CEF levels of Ni$_2$Mo$_3$O$_8$. First, the band top of low-energy spin waves is below 1.5 meV and the magnetic order is destroyed by an in-plane field of 8 T, indicating $J_1 \ll D_T \approx 1.2$ meV. In singlet-ground-state systems, one would expect a paramagnetic state at zero temperature, but the system orders below 6 K. Second, spin excitons and spin waves should only hybridize when they have similar energy scales, while excitons should be featureless in $Q$ if CEF levels have much higher energy than spin waves. In Ni$_2$Mo$_3$O$_8$, the excitonic bands at high energy (12-20 meV) are weakly dispersive in energy but show clear $Q$-dependence.

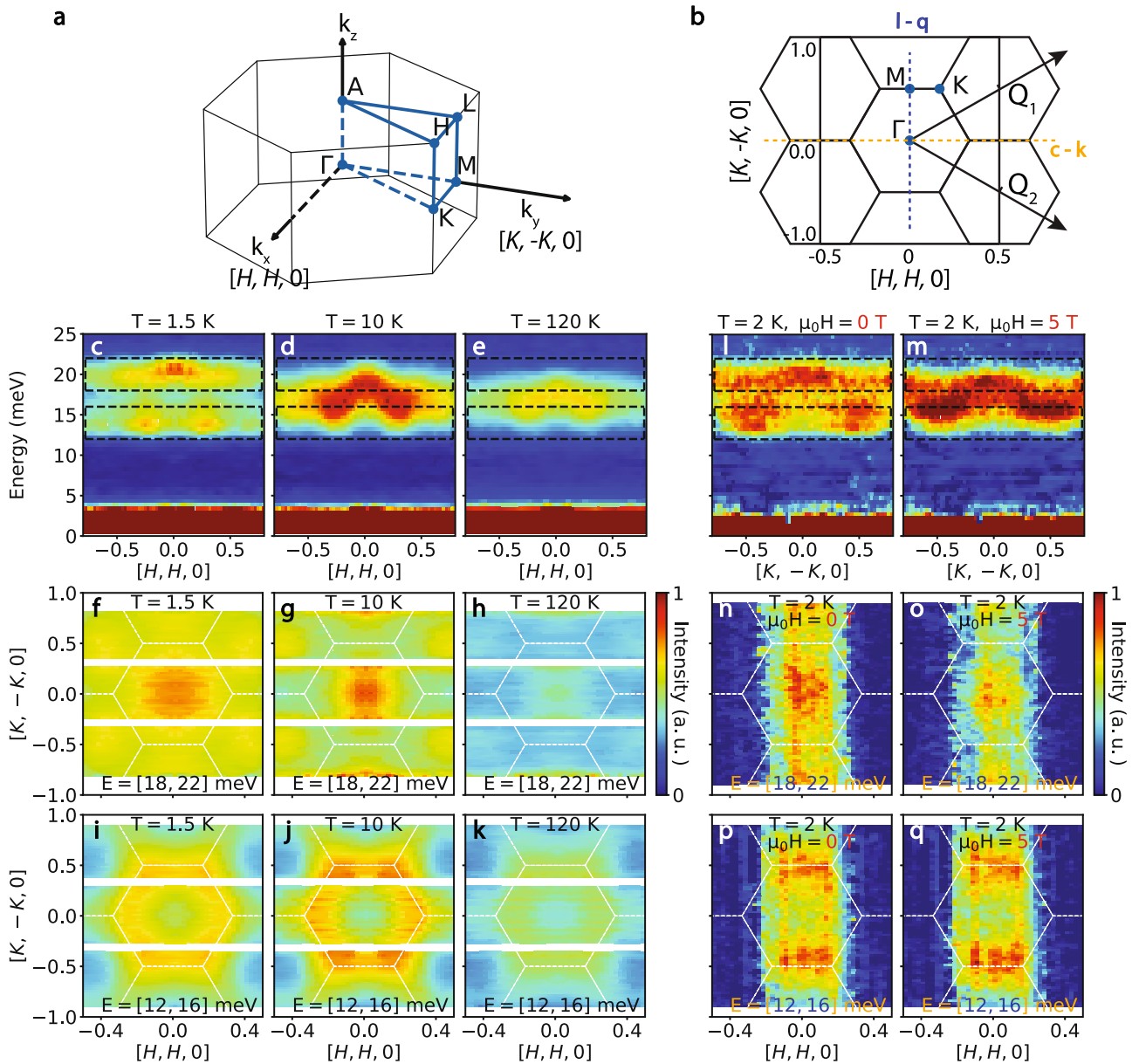

**Fig. 3 | Momentum and temperature dependence of magnetic scattering in Ni₂Mo₃O₈. a, b** 3D and 2D reciprocal space of $Ni_2Mo_3O_8$, where the high symmetry positions are marked. **c–e** Energy-momentum plots of the dispersions along the [H, H, 0] direction at 1.5, 10, and 120 K measured with 40 meV incident neutron energy. The data is integrated along the $L$ direction since the scattering above 12 meV has no modulation along the $L$ direction. Dashed lines indicate the energy integration range in **f–h**, Momentum dependence of the magnetic scattering at 1.5, 10, and 120 K, where high intensity is near the zone center. The energy integration range is 18–22 meV. **i–k** Momentum dependence of the scattering at 1.5, 10, and 120 K, where high intensity is near the zone boundary. The energy integration range is 12–16 meV. The scattering becomes sharper in the paramagnetic state at 10 K. **l–q** In-plane magnetic field dependence of $S(E, Q)$ at 0 and 5-T at 2 K. The missing data in h-q is due to the narrower detector coverage when $Ni_2Mo_3O_8$ is in a magnet.

To understand these results, we consider a $S = 1$ XXZ Hamiltonian for $Ni_2Mo_3O_8$:

$$H = H_1 + H_{23} + H_{SIA},\qquad(1)$$

where the first-nearest neighbor coupling is

$$H_1 = \sum_{i \in t} J_1(\mathbf{S}_i \cdot \mathbf{S}_{i+a} - \gamma(\mathbf{S}_i \cdot \hat{\mathbf{e}}_a)(\mathbf{S}_j \cdot \hat{\mathbf{e}}_a) + d(\hat{z} \times \hat{\mathbf{e}}_a) \cdot \mathbf{S}_i \times \mathbf{S}_{i+a}),\qquad(2)$$

the second- and third- neighbor coupling are

$$H_{23} = \sum_{\langle ij \rangle \in t} J_T \mathbf{S}_i \cdot \mathbf{S}_j + \sum_{\langle ij \rangle \in o} J_O \mathbf{S}_i \cdot \mathbf{S}_j + \sum_{\langle ij \rangle \in \{3NN\}} J_3 \mathbf{S}_i \cdot \mathbf{S}_j,\qquad(3)$$

and the SIA term is

$$H_{SIA} = \sum_{i \in t} D_T(S_i^z)^2 + \sum_{i \in o} D_O(S_i^z)^2.\qquad(4)$$

Here $\mathbf{S}_i$ is the spin operator at site $i$, $\gamma$ the anisotropic exchange, $\hat{\mathbf{e}}_a$ is the unit vector linking the spins at site $i$ and $i + a$, $D_T$ and $D_O$ are SIA at tetrahedral and octahedral sites, respectively. For $J = 0$, the Hamiltonian in Eq. (1) has a unique gapped ground state ($S_i^z = 0$ on all states). Perturbation for $J \ll D$ preserves the gap and the system remains in a unique, trivial ground state. For $J \gg D$, the single-ion terms are unimportant, and we expect an ordered ground state. Consequently, a quantum phase transition from paramagnetic to ordered state is

expected as a function of increasing $J$. We capture the transition by a Curie-Weiss mean-field approach (see Supplementary Information). At the simplest level, if we only consider nonzero $J_1, \gamma$ and $D_T, D_O$, the critical value of $J_1$ is found to be

$$\widetilde{J} = J_1(1 + \gamma/2) = \frac{\sqrt{D_T D_O}}{2}. \tag{5}$$

This equation shows that order can be induced when the exchange is intermediate between the two single-ion energies. In Ni$_2$Mo$_3$O$_8$, $D_T \approx 16$ meV and $D_O \approx 1$ meV, according to our CEF analysis. Therefore, the scale of magnetic exchange required to induce moment is largely reduced due to the bipartite nature of the lattice. From Eq. (5), we have $\widetilde{J} \approx 2.06$ meV, which is consistent with the low-energy scale of the magnetic order. The anisotropic exchange $\gamma$, which is originated from the combined effects of crystal field and SOC, can be used to energetically favor the experimentally observed four-sublattice state. The Dzyaloshinskii-Moriya interaction $d$ is responsible for the small out-of-plane spin canting[19], and a finite value above the threshold ($d \approx \widetilde{J}$) is necessary[19] to stabilize the stripy (as opposed to the Néel) ordered state that is observed experimentally.

The $\boldsymbol{Q}$-dependent scattering for Ni tetrahedral CEF levels in Fig. 3f, g and i, j suggest short-range ferromagnetic and AF correlations of Ni tetrahedrons, respectively. We first consider three tetrahedral Ni atoms on the vertices of an equilateral triangle and the spins forms a 120° configuration (Fig. 4a). The Fourier transform of the spins on this cluster is:

$$\boldsymbol{S}_c(h,k) = \sum_{j=0}^{2} \boldsymbol{S}_0 R_{1/3} F_{Ni}(|\boldsymbol{Q}_{hkl}|) \exp\left[-i2\pi\left(x_j h + x_y k\right)\right] = \boldsymbol{S}_0 \mathbb{M} F_{Ni}(|\boldsymbol{Q}_{hkl}|), \tag{6}$$

where $\boldsymbol{S}_0$ is the direction of one of the spins, $R_{1/3}$ is the 120° rotational matrix, $F_{Ni}$ is the magnetic form factor of Ni$^{2+}$, and $\mathbb{M}$ is the magnetization matrix. The observed scattering $S(E, \boldsymbol{Q})$ intensity is proportional to $|\hat{\kappa} \times (\boldsymbol{S}_c \times \hat{\kappa})|^2$, where $\hat{\kappa} \equiv \boldsymbol{Q}/Q$. Figure 4b shows the calculated

structural factor $S(\boldsymbol{Q})$ for the 120° AF spin configuration, where the $J_T$ in Ni tetrahedron triangles dominates. Similarly, the structural factor for a ferromagnetic spin configuration can be calculated by canceling the off-diagonal term of $\mathbb{M}$, which is shown in Fig. 4d. The calculated factors fit well with the observed spin excitations in the INS experiments, indicating that spin excitons have spin-spin correlations.

To quantitatively understand the $S(E, \boldsymbol{Q})$ spectrum of the CEF levels, we modeled the INS spectra using an effective Hamiltonian in Eqs. (1–5) to describe a ground state singlet and excited doublet on each octahedral and tetrahedral site with an effective $S = 1$ spins, SIAs, symmetric and anti-symmetric exchange interactions (see Supplementary Information). The large SIA on the tetrahedral site gives rise to the apparent high energy CEF modes, which disperse due to the exchange couplings. The spectral function of the excitations in the ordered state is calculated using a flavor wave expansion based on the SU(3) representation of the triplet of levels on each site. This method captures the partial suppression of ordered moment by the tendency to single-ion singlet formation, while still describing the ordering and associated spin waves.

Focusing on the high energy excitations, the flavor wave calculation predicts the formation of four bands, which are spin-split and folded due to the AF enlargement of the unit cell. The predominant momentum dependence arises from the AF exchange coupling between the closest pairs of tetrahedral sites ($J_T$), which leads to high intensity at the lower edge of the band near the zone boundary, and at the upper edge of the band near the zone center (right panels in Fig. 4b–d). The exchange parameters from the calculation are summarized in the Supplementary Information.

Our single crystal X-ray and neutron diffraction refinements find that Ni tetrahedrons in Ni$_2$Mo$_3$O$_8$ form an ideal 2D triangular lattice without the magnetic and nonmagnetic disorder (Table 1). As the energy scale of the CEF spin excitons from Ni tetrahedrons is much larger than the magnetic exchange interactions determined from spin waves of Ni$_2$Mo$_3$O$_8$ (Fig. 2a), the presence of static AF order only slightly modifies the continuum-like $\boldsymbol{Q}$-dependent scattering by making it less well-defined in the ordered state possibly due to mixing of the dispersive spin waves with CEF levels.

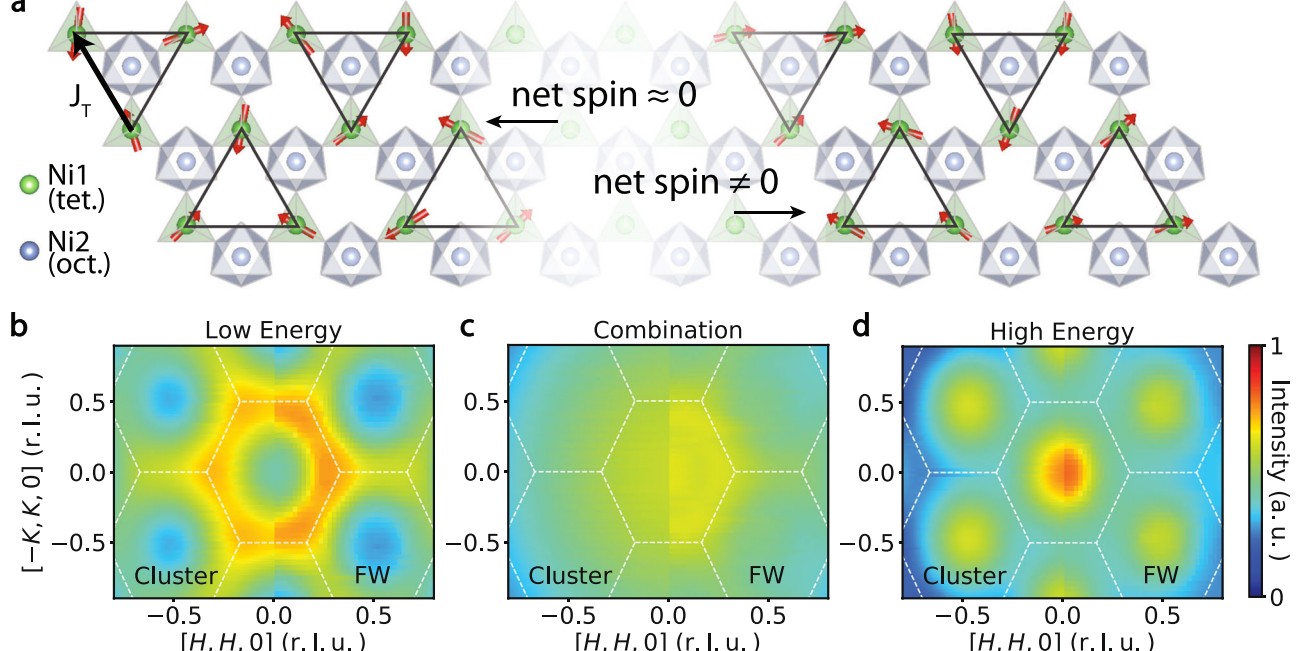

**Fig. 4 | Structural factors in Ni$_2$Mo$_3$O$_8$ calculated on spin clusters and by flavor wave (FW) method. a** Schematic of spin clusters used to calculate the structural factors. The 120° configurations with approximately zero net spin give rise to the pattern in which high intensity is near the zone boundary. The FM clusters lead to the pattern where high intensity is near the zone center. **b–d** Comparison of the structural factors by the cluster and flavor wave calculations. Both methods reproduce the scattering in Ni$_2$Mo$_3$O$_8$ at 12–16 meV (**b**), 18–22 meV (**d**), and the sum of two (**c**).

In induced-moment systems with a singlet-ground state[3–7], spin excitons can become highly dispersive and couple strongly to the ground state with a large magnetic exchange coupling $J$. When the ground state is not a spin singlet but a pseudospin doublet, spin excitons at high energies in some $d^8$ transition-metal oxides can also be dispersive and have unusual properties due to strong SOC. For example, in the classic Mott insulator CoO, where the strength of SOC is comparable to the magnetic exchange coupling[20], the $Q$-dependence of spin excitons at high energies decay faster with $Q$ than the $Co^{2+}$ magnetic form factor, suggesting a breakdown of the localized spin excitons towards spatially extended magnetism[21]. More recently, in $A$-type AF ordered CoTiO$_3$ with $T_N \approx 38$ K[22], the dispersive spin excitons around ~27 meV due to SOC become softer and acquire a larger bandwidth on warming from the AF (5 K) to the paramagnetic state (60–120 K)[23,24]. For comparison, the ~17 meV CEF doublet in the Ni tetrahedral site in Ni$_2$Mo$_3$O$_8$ induced by SOC (Fig. 2d) is similar in $Q$-space around the BZ boundary (see Supplementary Fig. 8 for the detailed cuts and comparisons) and narrower in energy bandwidth on warming from the AF (1.5 K) to the paramagnetic (10 and 120 K) state (Fig. 3c–k). While weakly dispersive spin waves above a large spin gap seen in the spin-chain compound Sr$_3$NiIrO$_6$[25] and the 1D magnet BaMo(PO$_4$)$_2$[26] also survive to temperatures well above their perspective $T_N$s, they originate from magnons (not spin excitons) and do not have the line-shapes in $Q$-space as we observe in Ni$_2$Mo$_3$O$_8$.

Therefore, our results highlight the novel physics associated with two magnetic species, each on a frustrated triangular lattice, with very different single-ion anisotropies, and expose Ni$_2$Mo$_3$O$_8$ as a promising venue to explore the propagation of spin excitons in a dense highly fluctuating magnetic background. Most importantly, they indicate that CEF levels in an ideal triangular lattice magnet can produce dispersive spin excitons irrespective of static magnetic order, and the origin of this phenomenon is most likely due to spin entanglement and geometric frustrations without invoking the quantum spin liquid paradigm[27–30].

## Methods

### Sample growth

Polycrystalline samples of Ni$_2$Mo$_3$O$_8$, NiZnMo$_3$O$_8$, and Zn$_2$Mo$_3$O$_8$ were synthesized using a solid-state method. Stoichiometric powders of NiO, ZnO, Mo, and MoO$_3$ were mixed and pressed into pellets, and then sintered at 1050 °C for 24 h. Single crystalline Ni$_2$Mo$_3$O$_8$ was synthesized using the chemical vapor transport method. Powder X-ray diffraction measurements performed on powder samples and ground single crystals reveal that the samples have a pure phase, with a space group $P6_3mc$ and the lattice parameter $a = b = 5.767$ Å and $c = 9.916$ Å. The structural information of the Ni$_2$Mo$_3$O$_8$ single crystal was investigated using a Rigaku XtaLAB PRO diffractometer equipped with a HyPi x-6000HE detector at ORNL. A molybdenum anode was used to generate X-rays with wavelength $\lambda = 0.7107$ Å. The samples were cooled by cold nitrogen flow provided by an Oxford N-Helix cryosystem. Single crystal X-ray refinements on Ni$_2$Mo$_3$O$_8$ reveal that the Ni and Mo are in fully occupied positions with no magnetic and nonmagnetic site disorder (Table 1).

### Specific heat measurements

Specific heat measurements were conducted using a thermal-relaxation method in a physical property measurement system (Quantum Design).

### Neutron diffraction

Neutron powder diffraction experiments were performed at room temperature using the high resolution powder diffractometer BT-1, at the NIST center for neutron research. 5.0 grams of NiZnMo$_3$O$_8$ powder was used. Powder neutron refinements reveal that Zn prefers to occupy tetrahedral sites (88.1%) and the rest of the tetrahedral sites are occupied by Ni (11.9%). Detailed results of the refinement are

shown in Supplementary Table 1. Single crystal neutron diffraction experiments were carried out using the elastic diffuse scattering spectrometer, CORELLI[31], at the Spallation Neutron Source, ORNL. One small piece of single crystalline Ni$_2$Mo$_3$O$_8$ was aligned in the [$H$, 0, $L$] plane. 53 structural Bragg peaks and 32 magnetic Bragg peaks at 2 K were used for the refinement.

### CEF level measurements on powder samples

INS experiments were carried out on polycrystalline Ni$_2$Mo$_3$O$_8$ (6.0 g), NiZnMo$_3$O$_8$ (4.24 g), and Zn$_2$Mo$_3$O$_8$ (4.33 g), on the chopper spectrometer, MARI, and the cold neutron multi-chopper spectrometer, LET, at ISIS neutron and muon source. We collected data with 40 meV, 250 meV, and 1.5 eV incident energy ($E_i$) at 4 K on MARI, and with 1.8, 3.7, and 12.1 meV $E_i$ at 2 and 12 K on LET.

### INS experiments on single crystals

We co-aligned more than 200 pieces of single crystals of Ni$_2$Mo$_3$O$_8$ (1.5 g) to carry out inelastic neutron experiments on the cold neutron chopper spectrometer, CNCS, the fine-resolution fermi chopper spectrometer[32], SEQUOIA[33] and ARCS[34] thermal chopper spectrometers, at the Spallation Neutron Source, Oak Ridge National Laboratory. The sample assembly was aligned in [$H$, $K$, 0] scattering plane on CNCS and SEQUOIA. We performed 180° rotational scans at 1.7, 3.5, 4.5, 5.5, and 6.5 K with 2.5 meV $E_i$ on CNCS and at 1.5, 10, and 120 K with 40 meV E$_i$ on SEQUOIA. On ARCS, we aligned the sample in [$-K$, $K$, $L$] scattering plane and measured with 26 and 40 meV $E_i$ at 2 K.

### Flavor wave calculations

Each mean-field state of the quantum magnet corresponds to a product state where the wave function on each site lives in the spin-1 Hilbert space. One can design a trial wave function to describe the magnetic order, with the variational parameters determined by minimization of the mean-field energy. The excitations on top of the ground state can then be suitably described by the "flavor waves" making use of the SU(3) flavor rotation in the spin-1 Hilbert space (see Supplementary Information for details). The dispersive excitations are well-defined in both ordered and disordered phases and can exhibit the experimentally observed behaviors in structure factors. In Fig. 4, we have used the parameters

$$D_t = 16 \text{ meV}, D_o = 1.0 \text{ meV}, d = 0.3, \gamma = 0.5,$$

$$J_1 = 2.0 \text{ meV}, J_t = 0.5 \text{ meV}, J_o = 0.15 \text{ meV}, J_3 = 0.0 \text{ meV}.$$

## Data availability

The data that support the plots in this paper and other findings of this study are available from the corresponding author on reasonable request.

## Code availability

The computer codes used to calculate flavor waves in this study are available from the corresponding author on reasonable request.

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

## Acknowledgements

We thank Haoyu Hu and Youzhe Chen for helpful discussions. The INS work at Rice is supported by the U.S. DOE, BES under grant no. DE-SC0012311 (P.D.). The single crystal growth and characterization at Rice are supported by the Robert A. Welch Foundation Grant No. C-1839 (P.D.). S.L. and A.H.N. were supported by the Robert A. Welch Foundation Grant No. C-1818. C.L.H. and E.M. acknowledge support from the Robert A. Welch Foundation Grant No. C-2114. L.B. was supported by the NSF CMMT program under Grant No. DMR-2116515. C.L.H. is also supported by the Ministry of Science and Technology (MOST) in Taiwan under grant No. MOST 109-2112-M-006-026-MY3 and MOST 110-2124-M-006-009. A portion of this research used resources at the Spallation Neutron Source, a DOE Office of Science User Facility operated by ORNL. Experiments at the ISIS Neutron and Muon Source were supported by a beam time allocation RB2090032 and RB2090034 from the Science and Technology Facilities Council. We thank Ross Stewart for performing the INS experiment on the powder samples using LET.

## Author contributions

P.D. and B.G. conceived the project. B.G. and T.C. made equal contributions to the project. B.G., T.C., and J.L. prepared the samples. Heat capacity measurements were performed by C.L.H. in E.M. lab. The crystal field measurements and analysis were performed by B.G., T.C., J.R.S., M.D.L., and D.T.A. Single crystal neutron and X-ray diffraction experiments were carried out and analyzed by B.G., T.C., and F.Y. INS experiments were carried out and analyzed by B.G., T.C., J.L., M.B.S., A.P., and D.L.A. The spin-wave analysis was done by S.L., L.C., and A.H.N. The flavor wave analysis was done by X.C.W., M.F., and L.B. The manuscript is written by B.G., T.C., L.B., and P.D. All authors made comments.

## Competing interests

The authors declare no competing interests.
