## [Peer Review File · Nature Communications]

Reviewers' Comments:

Reviewer #1:

Remarks to the Author:

Following my previous report, I find this manuscript quite interesting and feel generally supportive of its publication in Nature Comm. It is a carefully done work that reveals spin excitons and associated excitation continuum in a model triangular antiferromagnet. However, I would like to raise several points that should be addressed before publication:

1. I deem the new title unnecessarily arcane and quite difficult for a general reader. I suggest that the authors reconsider the earlier title ("Diffusive excitonic bands...") as a more intuitive one. Moreover, it highlights the triangular geometry of the material.
2. I do not quite share the authors' view that their work relates to the excitation continua of quantum spin liquids (QSL). In the year 2022, one would hardly associate every excitation continuum with a QSL. Especially in triangular antiferromagnets magnon breakdown leading to the continua is well-known and has been documented in systems like Ba₃CoSb₂O₉ [Nature Comm. 8, 235 (2017)] and LuMnO₃ [PRL 111, 257202 (2013)]. In my opinion, repeated mentions of the QSL in both abstract and introduction are unnecessary and even misleading, because the rest of the manuscript has nothing to do with a QSL. I think that concentrating on spin excitons would be more useful in the introduction.
3. I still do not understand why the authors avoid describing all excitations at 1.5 K using spin-wave theory. A model with the large single-ion anisotropy should account for the excitations not only at 1-2 meV but also at 10-20 meV. Previous studies of BaMo(PO₄)₂ and Sr₃NiIrO₆ (see references in my previous report) were very successful in describing spin excitons via spin-wave theory, and I do not see why this approach can not be used in the present work. This analysis could help one to answer the principle question whether the excitations at 10-20 meV form a continuum even at 1.5 K, or they can be treated as magnons of the model with the large single-ion anisotropy.
4. A moderate exchange coupling J_1 is indeed sufficient to induce magnetic order on the tetrahedral sites. However, I do not understand why the ordered moment on the tetrahedral site is higher than on the octahedral one, even though the tetrahedral site experiences condensation. A similar situation arises in magnets with dimer units where individual dimers should be in the nonmagnetic singlet state, but interactions to nondimerized atoms induce long-range order with a finite ordered moment also on the dimer sites [for example: PRB 80, 104405 (2009)]. However, in that case the ordered moment on the dimer sites is significantly reduced. Why does not it happen in Ni₂Mo₃O₈? Again, spin-wave calculations would be useful in order to clarify this issue.

Reviewer #2:

Remarks to the Author:

I would like to thank the authors for considering my comments and modifying the manuscript to address some of the issues. In the revised manuscript, the authors are emphasizing that their work demonstrates that "the presence of a continuum is a necessary but not a sufficient condition for the fractionalization or long-range entanglement of quantum spins". This statement is correct and not controversial (I don't know if anyone claimed that it is a sufficient condition...), but I am not sure if the current work "demonstrates" this. The main claim here is the observation of a spin excitation continuum in a non-QSL system, but I still think that the observed spin-exciton spectra

is better described as a dispersion relation rather than a continuum. In my earlier report, I expressed concerns about the rather well-defined nature of the observed excitations. The authors wrote "The correlations are thus short-ranged and similar with the ones observed in 3D QSL candidates..." which addresses the concern about the momentum-space, but does not directly answer the concern about the energy bandwidth. As an extreme example, local CEF excitations are well defined (narrow-bandwidth) in energy but 'continuum' in momentum space because there is no k -dependence. No one will describe this as a continuum. As described in the revised manuscript, the spin-exciton excitations become dispersive due to magnetic correlation, giving rise to a finite energy bandwidth, which is similar to what was observed in some other systems. The authors however claims that the spin-exciton in $\text{Ni}_2\text{Mo}_3\text{O}_8$ is different because it is "sharper in Q -space around the BZ boundary and narrower in energy bandwidth on warming from the AF (1.5 K) to the paramagnetic (10 K and 120 K) state (Figs. 3c-3k)". It is not obvious to me that features are sharper in Fig. 3h and Fig. 3k based on the data presented here. Even if they are, I have hard time believing such a subtle difference merits a whole new interpretation. In summary, I do not agree with the interpretation of the high energy excitation as a continuum. Therefore, I do not recommend the current manuscript in the current form for publication in Nature Communications.

Reviewer #1 (Remarks to the Author):

Following my previous report, I find this manuscript quite interesting and feel generally supportive of its publication in Nature Comm. It is a carefully done work that reveals spin excitons and associated excitation continuum in a model triangular antiferromagnet. However, I would like to raise several points that should be addressed before publication:

1. I deem the new title unnecessarily arcane and quite difficult for a general reader. I suggest that the authors reconsider the earlier title ("Diffusive excitonic bands...") as a more intuitive one. Moreover, it highlights the triangular geometry of the material.

We appreciate the reviewer's comments and change the title back to the original one.

2. I do not quite share the authors' view that their work relates to the excitation continua of quantum spin liquids (QSL). In the year 2022, one would hardly associate every excitation continuum with a QSL. Especially in triangular antiferromagnets magnon breakdown leading to the continua is well-known and has been documented in systems like Ba₃CoSb₂O₉ [Nature Comm. 8, 235 (2017)] and LuMnO₃ [PRL 111, 257202 (2013)]. In my opinion, repeated mentions of the QSL in both abstract and introduction are unnecessary and even misleading, because the rest of the manuscript has nothing to do with a QSL. I think that concentrating on spin excitons would be more useful in the introduction.

We appreciate the reviewer's comments. As mentioned by the reviewer, the spin excitation continuum features are seen from magnons in Ba₃CoSb₂O₉ and LuMnO₃. The magnetic excitations seen in Ni₂Mo₃O₈ in the energy range of [12, 22] meV are from CEF (not magnons), and can be observed in both AF and paramagnetic states, fundamentally different from Ba₃CoSb₂O₉ and LuMnO₃.

To address the referee's concern, we added several lines in the paper to discuss other situations where non-spin wave excitations are seen, as discussed by the referee and cited the above mentioned papers. We also described the CEF scattering in the range of [12, 16] meV (Fig. 3i-3k) as a diffusive pattern around BZ boundaries in moment space instead of a continuum. Such pattern is widely used as strong evidence for a QSL [PRB 100.14 144432 (2019), PRB 100.22 220407 (2019), PRX 11.2 021044 (2021)]. We felt that QSL discussion is still one of the major motivations of the work, and for this reason, we decided to keep the discussion there. We hope that the referee will agree with us.

3. I still do not understand why the authors avoid describing all excitations at 1.5 K using spin-wave theory. A model with the large single-ion anisotropy should account for the excitations not only at 1-2 meV but also at 10-20 meV. Previous studies of BaMo(PO₄)₂ and Sr₃NiIrO₆ (see references in my previous report) were very successful in describing spin excitons via spin-wave theory, and I do not see why this approach can not be used in the present work. This analysis could help one to answer the principle question whether the excitations at 10-20 meV form a

continuum even at 1.5 K, or they can be treated as magnons of the model with the large single-ion anisotropy.

We appreciated the reviewer's comment. As discussed in our previous replies to the reviewer, the reason that spin excitations in above mentioned references survive well above T_N is because the 1D nature of the magnetic interactions, which is clearly different from the triangular lattice system addressed here. In principle, the referee is correct that one can try to assume that spin excitation seen in 12-20 meV are spin waves instead of CEF levels as we discussed in the paper. However, this would require a much larger spin anisotropy gap and a higher magnetic ordering temperature. Flavour wave theory is a generalized version of spin-wave theory and has better performance in describing a spin-1 system. It can also capture the dispersive feature of the scattering in 12-20 meV at the same time.

In fact, we did use linear spin wave theory (LSWT) to fit the 12-20 meV data at 1.5 K first, but simply cannot. In the two references mentioned by the reviewer, there is only one branch of spin waves in both calculation and experimental data. Of course, one can fit just one band using spin-wave theory. To fit both branches and understand their energy splitting below T_N , one will need to build up a spin wave model with very large anisotropy gap between the two modes, which is very difficult and disagrees with magnetic ordering temperature/thermodynamic measurements.

From the low energy range INS data (Fig 2a & Fig S9), we can see that there are two branches of spin waves (acoustic and optical modes) at energies below 2 meV, which is all the spin waves in the series of compounds $M_2Mo_3O_8$ ($M=Mn, Fe, Co, Ni$) since there are only two magnetic ions per unit cell. One example is the spin waves in $Co_2Mo_3O_8$ from Ref 38 and there cannot be more than 2 modes from LSWT calculation. In the case of $Ni_2Mo_3O_8$, there are at least two levels in the energy range of [12,22] meV (more if we consider splitting below T_N). If we count these two as spin waves, there would be 4 modes in the spin waves if we include spin wave data at lower energies (below 2 meV). This is impossible and contradicts to the LSWT prediction. This leaves only one option for the origin of the excitations in the energy range of [12,22] meV, i.e, that they are from CEF of Ni and not spin waves. We revised the manuscript to make this point clear.

Another point is the comparison with the isostructural compound $Co_2Mo_3O_8$, which has a $T_N = 40$ K, and an upper branch of spin wave around 12 meV (Ref 38). $Ni_2Mo_3O_8$ has a $T_N = 5.5$ K, almost an order of magnitude smaller than $Co_2Mo_3O_8$. The energy range of spin wave in $Ni_2Mo_3O_8$ should be a magnitude smaller too, which is around 1 meV, consistent with our experimental data.

4. A moderate exchange coupling J_1 is indeed sufficient to induce magnetic order on the tetrahedral sites. However, I do not understand why the ordered moment on the tetrahedral site is higher than on the octahedral one, even though the tetrahedral site experiences condensation. A similar situation arises in magnets with dimer units where individual dimers should be in the nonmagnetic singlet state, but interactions to nondimerized atoms induce long-range order with a finite ordered moment also on the dimer sites [for example: PRB 80, 104405 (2009)]. However, in that case the ordered moment on the dimer sites is significantly reduced. Why does not it happen in $Ni_2Mo_3O_8$? Again, spin-wave calculations would be useful in order to clarify this

issue.

We appreciated the reviewer's comment. We are not sure what the reviewer means by "even though the tetrahedral site experiences condensation". The tetrahedral site and octahedral site both have a singlet ground state and a doublet first excited state, as seen in our schematic CEF analysis in the SI. The only difference is the energy range of 1st excited state. They are different from the case (dimer vs non-dimer) in the reference mentioned by the reviewer.

The reason for the tetrahedral site Ni²⁺ having higher moment is that the *g* factor on the tetrahedral site is higher. From the previous ESR measurements (Ref 24), "the ratio of the tetrahedral site *g* factor to the octahedral site *g* factor determined by ESR at T = 290 K is 1.46", and tetrahedral site *g* factor increases a bit as T decreases. This ratio is very close to the ratio of the refined moments of tetrahedral and octahedral sites.

In the further revised draft, we cut down the discussion on QSL in the introduction, and discussed more why excitations at 10-20 meV cannot be due to spin waves of Ni. With these changes, we hope that the reviewer will agree that the paper is suitable for publication.

Reviewer #2 (Remarks to the Author):

I would like to thank the authors for considering my comments and modifying the manuscript to address some of the issues. In the revised manuscript, the authors are emphasizing that their work demonstrates that "the presence of a continuum is a necessary but not a sufficient condition for the fractionalization or long-range entanglement of quantum spins". This statement is correct and not controversial (I don't know if anyone claimed that it is a sufficient condition...), but I am not sure if the current work "demonstrates" this. The main claim here is the observation of a spin excitation continuum in a non-QSL system, but I still think that the observed spin-exciton spectra is better described as a dispersion relation rather than a continuum. In my earlier report, I expressed concerns about the rather well-defined nature of the observed excitations. The authors wrote "The correlations are thus short-ranged and similar with the ones observed in 3D QSL candidates..."

which addresses the concern about the momentum-space, but does not directly answer the concern about the energy bandwidth. As an extreme example, local CEF excitations are well defined (narrow-bandwidth) in energy but 'continuum' in momentum space because there is no *k*-dependence. No one will describe this as a continuum. As described in the revised manuscript, the spin-exciton excitations become dispersive due to magnetic correlation, giving rise to a finite energy bandwidth, which is similar to what was observed in some other systems. The authors however claims that the spin-exciton in Ni₂Mo₃O₈ is different because it is "sharper in *Q*-space around the BZ boundary and narrower in energy bandwidth on warming from the AF (1.5 K) to the paramagnetic (10 K and 120 K) state (Figs. 3c-3k)". It is not obvious to me that features are sharper in Fig. 3h and Fig. 3k based on the data presented here. Even if they are, I have hard time believing such a subtle difference merits a whole new interpretation.

In summary, I do not agree with the interpretation of the high energy excitation as a continuum.

Therefore, I do not recommend the current manuscript in the current form for publication in Nature Communications.

We appreciate the reviewer's comment.

In the latest draft, we described the scattering as diffuse CEF excitations / excitons in the range of [12, 16] meV (Fig. 3i-3k) as a diffusive pattern around BZ boundaries in moment space instead of a continuum, which should eliminate the confusion about the "continuum".

Again, the excitations we are talking about here is not as board in energy and momentum space as in the 1D chain materials such as KCuF3. We are comparing the excitations in [12-16 meV] in Ni2Mo3O8 to that of more general 2D and 3D QSL candidates, not the 1D case.

For the concerns about the energy bandwidth, the continuum in Ni2Mo3O8 is also similar to those in 2D and 3D QSL candidates, such as Ca10Cr7O28, PbCuTe2O6, and triangular lattice magnet NaYbSe2, which is about 1 ~ 2 meV.

As mentioned in previous replies, the correlation lengths of the scattering are around 5 ~10 Å. They are short-ranged and similar with the ones observed in other 2D and 3D QSL candidates. Therefore, the excitations in Ni2Mo3O8 are similar to those in 2D and 3D QSL candidates.

We would not describe the extreme example of local CEF excitation without k-dependence as continuum. The spin exciton is "narrower in energy bandwidth", which is obvious from Fig 2c. "Sharper in Q-space around the BZ boundary" is from the cuts and the corresponding FWHM in the AF state and paramagnetic state in Fig S8 in the SI. We revised the manuscript to make clear how we get the feature. The key point is that although dispersive spin excitons have been reported before, the continuum around the BZ boundary in spin exciton (not magnon) has not been observed before. We give an interpretation to this new feature, not a new interpretation to a feature with subtle differences.

Reviewers' Comments:

Reviewer #1:

Remarks to the Author:

I would like to thank the authors for their detailed response, especially on the description of excitations with linear spin-wave theory. I see that the main criticism, both from my side and from the second reviewer, revolves around the question whether diffusive features should be called a continuum, and whether its observation has any implications to interpreting spinon continuum of a QSL. In this respect, I find the last sentence of the revised abstract very controversial. First, it is grammatically inconsistent ("geometric frustration... can have diffusive CEF excitations" - how does frustration have excitations?), and second, it still evokes the idea that the observation of an excitation continuum is not a sufficient condition for the quantum spin liquid. This idea is certainly not new, and in 2022 it can't be the main conclusion of a manuscript published in Nature Communications. Systems with lattice disorder, like YbMgGaO₄, already show quite prominently that the continuum does not always indicate a QSL. Any further demonstration is a useful addition to that, but it is not a breakthrough result with significant impact.

I stay with the opinion that the more interesting aspect of this work belongs to the dispersion of spin excitons, even in the absence of long-range magnetic order. For me the most nontrivial and insightful part here is the distinct Q-dependence of the spectral weight in different parts of the excitonic bands (for example, panels 'g' and 'j' of Fig. 3) and the possibility of describing it using different spin configurations on the triangles (Fig. 4). My understanding of this result is that the excitonic "continuum" does actually arise from some sort of entanglement between the spins, and this entanglement can be tracked by measuring spin excitons.

Overall, I readily agree with the concluding remarks on page 15 ("...our results highlight the novel physics... and expose Ni₂Mo₃O₈ as a promising venue to explore the propagation of spin excitons... CEF levels can produce QSL-like spin excitation continuum... most likely due to geometric frustration"), but I do not think that these ideas are adequately reflected in the abstract, which sounds much more trivial than the result actually is. I remain strongly supportive of publication in Nature Comm., but my recommendation, like in the previous round, would be a stronger focus on presenting the physics of spin excitons instead of reiterating the continuum vs. QSL relation.

Reviewer #2:

Remarks to the Author:

I would like to thank the authors for addressing some of the concerns expressed in my previous report, such as using "diffusive" to describe the spin-excitons. However, I do not think this simple language change is sufficient to alleviate my concerns.

First, I do not agree with the authors' claim that the spin exciton is "shaper in Q-space around the BZ boundary... on warming..." If one examines Fig S8 in the SI carefully, it is easy to see that the experimental data are quite similar at the two temperatures. It all depends on what kind of background is used to fit the data. In other words, one cannot make this statement based on Fig. S8, if experimental uncertainty is considered.

This brings up the underlying issue of how the data is presented in this paper. The authors seem to be preoccupied with QSL phenomenology and they are trying to somehow connect their data to QSL motivation. I do not think that's very useful for the readers and maybe even confusing.

Lastly, I have minor comments about the things I noticed this time (not sure if this was the issue before).

- I do not understand why the authors use "moment space" instead of momentum space or Q-space. In this paper, "moment" is used to refer to the magnetic moment, so this usage is very confusing.

- Fig. S8 data in (a) and (c) should be 1.5 K, not 1.7 K, if they are from Fig. 3.

Replies to reviewer #1.

"I would like to thank the authors for their detailed response, especially on the description of excitations with linear spin-wave theory. I see that the main criticism, both from my side and from the second reviewer, revolves around the question whether diffusive features should be called a continuum, and whether its observation has any implications to interpreting spinon continuum of a QSL. In this respect, I find the last sentence of the revised abstract very controversial. First, it is grammatically inconsistent ("geometric frustration... can have diffusive CEF excitations" - how does frustration have excitations?), and second, it still evokes the idea that the observation of an excitation continuum is not a sufficient condition for the quantum spin liquid. This idea is certainly not new, and in 2022 it can't be the main conclusion of a manuscript published in Nature Communications. Systems with lattice disorder, like YbMgGaO₄, already show quite prominently that the continuum does not always indicate a QSL. Any further demonstration is a useful addition to that, but it is not a breakthrough result with significant impact."

We appreciate very much these comments from the reviewer that the continuum does not always indicate a QSL is already reported in YbMgGaO₄. After thinking through carefully, we further agree with the reviewer's conclusion that "Any further demonstration is a useful addition to that, but it is not a breakthrough result with significant impact". Our original thought was that the key difference between Ni₂Mo₃O₈ and YbMgGaO₄ is that the former does not have any magnetic and nonmagnetic site disorder, which is believed to cause the observed spin excitation continuum. Nevertheless, we believe that key conclusions of our work suggested by the reviewer is indeed more significant, and we have therefore completely rewritten the introduction and abstraction to reflect these conclusions. We believe that the reviewing process really improved the readability and highlighted the importance of the work not recognized in our original draft, we appreciate very much the suggestions from the reviewer.

"I stay with the opinion that the more interesting aspect of this work belongs to the dispersion of spin excitons, even in the absence of long-range magnetic order. For me the most nontrivial and insightful part here is the distinct Q-dependence of the spectral weight in different parts of the excitonic bands (for example, panels 'g' and 'j' of Fig. 3) and the possibility of describing it using different spin configurations on the triangles (Fig. 4). My understanding of this result is that the excitonic "continuum" does actually arise from some sort of entanglement between the spins, and this entanglement can be tracked by measuring spin excitons."

Again, we appreciate these comments from the reviewer and agree with them. Indeed, it is quite novel to have dispersion in spin excitons even in the absence of long range magnetic order. In the revised draft, we rewrote the introduction to reflect the changes suggested by the referee. As we mentioned above, we appreciate these very insightful comments aimed at highlighting the key discovery of the paper. With these changes, we feel that the paper represents a significant advancement suitable for Nature Communications.

“Overall, I readily agree with the concluding remarks on page 15 (“...our results highlight the novel physics... and expose Ni₂Mo₃O₈ as a promising venue to explore the propagation of spin excitons... CEF levels can produce QSL-like spin excitation continuum... most likely due to geometric frustration”), but I do not think that these ideas are adequately reflected in the abstract, which sounds much more trivial than the result actually is. I remain strongly supportive of publication in Nature Comm., but my recommendation, like in the previous round, would be a stronger focus on presenting the physics of spin excitons instead of reiterating the continuum vs. QSL relation.”

As discussed in replies above, we have revised the abstract and introduction of the paper to follow the recommendation of the referee. We hope that the referee will agree that the changes made improved the manuscript sufficiently to warrant its publication in Nature Communications.

Replies to reviewer #2.

“I would like to thank the authors for addressing some of the concerns expressed in my previous report, such as using “diffusive” to describe the spin-excitons. However, I do not think this simple language change is sufficient to alleviate my concerns.”

We appreciate very much these comments and agree with them. The referee raised the same issue as that of the referee 1 as discussed in his/her comments below. In the revised draft, we address these concerns and hope that the referee will agree changes made are sufficient to justify publication of this work.

“First, I do not agree with the authors’ claim that the spin exciton is “shaper in Q-space around the BZ boundary... on warming...” If one examines Fig S8 in the SI carefully, it is easy to see that the experimental data are quite similar at the two temperatures. It all depends on what kind of background is used to fit the data. In other words, one cannot make this statement based on Fig. S8, if experimental uncertainty is considered.”

Again, we agree with the referee that there are insufficient statistics to claim a significant sharpening in Q on warming from antiferromagnetic ordered state to paramagnetic state. We have revised the paper to reflect this point. Since this is not a major point of the paper and does not change the conclusion of the work, we hope that the referee will agree with changes made to the manuscript.

“This brings up the underlying issue of how the data is presented in this paper. The authors seem to be preoccupied with QSL phenomenology and they are trying to somehow connect their data to QSL motivation. I do not think that’s very useful for the readers and maybe even confusing.”

These comments are similar to the comments of the referee 1, and we appreciate them very much. The key messages of the paper are revised, following the suggestions of referees. We trust that the revised paper has addressed the concerns of the referee.

“Lastly, I have minor comments about the things I noticed this time (not sure if this was the issue before).

- I do not understand why the authors use “moment space” instead of momentum space or Q-space. In this paper, “moment” is used to refer to the magnetic moment, so this usage is very confusing.

- Fig. S8 data in (a) and (c) should be 1.5 K, not 1.7 K, if they are from Fig. 3.”

Thank you very much for noticing these typos, they have been corrected in the revised draft.